RESEARCH ARTICLES

# Social influences on delayed gratification in New Caledonian crows and Eurasian jays

**Rachael Miller**[1,2‡*], **James R. Davies**[2‡*], **Martina Schiestl**[3], **Elias Garcia-Pelegrin**[4], **Russell D. Gray**[5,6], **Alex H. Taylor**[5,7,8,9], **Nicola S. Clayton**[2]

**1** School of Life Sciences, Anglia Ruskin University, Cambridge, United Kingdom, **2** Department of Psychology, University of Cambridge, Cambridge, United Kingdom, **3** Faculty for Veterinary Medicine, University of Veterinary Science, Brno, South Moravia, Czech Republic, **4** Department of Psychology, National University of Singapore, Singapore, Singapore, **5** ICREA, Barcelona, Spain, **6** Max Planck Institute for the Science of Human History, Max Planck Society, Jena, Germany, **7** Institut de Neurociències, Universitat Autònoma de Barcelona, Barcelona, Spain, **8** School of Biological Sciences, University of Canterbury, Christchurch, New Zealand, **9** School of Psychology, The University of Auckland, Auckland, New Zealand

‡ RM and JRD are joint first authors on this work.
\* rh87@aru.ac.uk (RM); jd940@cam.ac.uk (JRD)

**Data Availability Statement:** The full data set and R script is available on Figshare. DOI: 10.6084/m9.figshare.23514828 Direct link: https://figshare.com/s/3a6adfae2cb31f707659

**Funding:** The study was funded by a Research Support Development Grant and Biology QR funds (Anglia Ruskin University) awarded to R.M, a Royal

## Abstract

Self-control underlies goal-directed behaviour in humans and other animals. Delayed gratification - a measure of self-control - requires the ability to tolerate delays and/or invest more effort to obtain a reward of higher value over one of lower value, such as food or mates. Social context, in particular, the presence of competitors, may influence delayed gratification. We adapted the 'rotating-tray' paradigm, where subjects need to forgo an immediate, lower-quality (i.e. less preferred) reward for a delayed, higher-quality (i.e. more preferred) one, to test social influences on delayed gratification in two corvid species: New Caledonian crows and Eurasian jays. We compared choices for immediate vs. delayed rewards while alone, in the presence of a competitive conspecific and in the presence of a non-competitive conspecific. We predicted that, given the increased risk of losing a reward with a competitor present, both species would similarly, flexibly alter their choices in the presence of a conspecific compared to when alone. We found that species differed: jays were more likely to select the immediate, less preferred reward than the crows. We also found that jays were more likely to select the immediate, less preferred reward when a competitor or non-competitor was present than when alone, or when a competitor was present compared to a non-competitor, while the crows selected the delayed, highly preferred reward irrespective of social presence. We discuss our findings in relation to species differences in socio-ecological factors related to adult sociality and food-caching (storing). New Caledonian crows are more socially tolerant and moderate cachers, while Eurasian jays are highly territorial and intense cachers that may have evolved under the social context of cache pilfering and cache protection strategies. Therefore, flexibility (or inflexibility) in delay of gratification under different social contexts may relate to the species' social tolerance and related risk of competition.

Society of New Zealand Rutherford Discovery Fellowship and a Prime Minister's McDiarmid Emerging Scientist Prize to A.H.T., and by the European Research Council under the European Union's Seventh Framework Programme (FP7/2007-2013)/ERC Grant Agreement No. 3399933, awarded to N.S.C. The funders had no role in study design, data collection and analysis, decision to publish, or preparation of the manuscript.

**Competing interests:** The authors have declared that no competing interests exist.

## Introduction

Self-control underlies decision-making and future planning, ensuring individuals are able to perform goal-directed behaviours. This process is important for humans and other animals [1, 2]. Self-control is influenced by socio-environmental factors in humans. For instance, it correlates with behavioural problems like substance abuse [3], and with measures of success, like social and academic competence [4]. It is also influenced by socio-environmental factors in other animals, such as sociality [5]. One measure of self-control is the ability to delay gratification, i.e. to tolerate a delay and/or invest more effort to obtain a reward of higher value over one of lower value, such as food or mates [6]. It has been tested comprehensively using various paradigms in many species, including primates and birds [7–13]. For instance, in the exchange paradigm, subjects may choose to swap rewards with a conspecific or experimenter for a more preferred reward [14].

However, the role of social context on self-control is still relatively unexplored. In humans, the presence and behaviour of others can influence our own decisions [15]. For example, children engage higher cognitive control when competing or cooperating with another person [16] and are less likely to delay gratification when the experimenter behaves in an unreliable/untrustworthy manner [17]. Flexibility in self-control is likely to be important in a social context in non-human animals too, for instance, refraining from approaching food or a potential mate while in the presence of a competitor [18, 19]. There are few delayed gratification studies that require interaction and co-operation with a conspecific, mostly using the token-exchange paradigm in primates [20, 21]. For example, high-ranking capuchin monkeys quickly acquired token exchange behaviour in social contexts, though low-ranking ones did not display this behavior [22]. There is therefore scope for developing tasks that explore the influence of social context and the behaviour of others on self-control.

Corvids (members of the crow family) have been found to differ in their ability to delay gratification [10, 23]. Corvids differ in sociality, i.e. living in a variety of different social systems [24]. For example, some corvids, such as Eurasian jays (E jays: *Garrulus glandarius*), are most often found alone or within a (socially) monogamous pair, who fiercely protect their own individual territories [24]. At the other extreme are the highly social corvids, such as rooks (*Corvus frugilegus*) and Western jackdaws (*Coloeus monedula*), who form large aggregations of up to 60,000 individuals [24], in which there can be a strong social hierarchy and colonial breeding [25]. Other species, such as New Caledonian crows (NC crows: *Corvus moneduloides*), common ravens (*Corvus corax*) and carrion crows (*Corvus corone*), show more flexibility in their sociality depending on season and age [26]. They sometimes remain within mating pairs or otherwise form larger family groups with overlapping territories and even showing some instances of cooperative breeding [24].

Studies suggest that corvids possess complex cognitive abilities, such as the ability to plan for the future [27, 28], mentally represent problems [29, 30], make inferences [31–33], and learn abstract information [34, 35]. In the social domain, corvids show evidence for co-operative behaviors [36–38] and seem to be aware of what other individuals can see and flexibly adjust their behaviour in response. For example, ravens differentiate between knowledgeable and ignorant conspecifics [39] even after controlling for observable behavioural cues [19, 40]. Furthermore, Western scrub jays (*Aphelocoma californica*) re-cache their food if they have been observed by a potential pilferer during caching, but not after caching in private [41] or when observed by their mate [42]. Importantly, this re-caching only occurs when the caching jays have themselves had experience of pilfering other individuals' caches [41].

Western scrub jays (*Aphelocoma californica*) are able to keep track of which birds were watching them during caching, as they only defend caches against subordinates and are tolerant to their partner sharing food [42]. Like scrub-jays, Eurasian jays have also demonstrated the use and flexible deployment of various cache-protection strategies [43–46] (although see [47]). Jays cached more behind an occluder [43] and at a distance [44] when observed by a conspecific than when alone, preferentially cached less in a 'noisy' substrate when a conspecific could hear but not see them (but not when they could hear and see them) [45].

There is variation across species in the socio-cognitive abilities of corvids. Some evidence suggests that these abilities vary with the species' natural sociality. For example, when comparing highly social pinyon jays (*Gymnorhinus cyanocephalus*) with less social Western scrub-jays on two complex tasks related to tracking and assessing social relationships, the pinyon jays learned more rapidly and were significantly more accurate than the scrub jays [48]. Additionally, the ability to remember the locations of conspecific made caches (observation spatial memory) in order to take them later, seems to vary in line with a species' sociality, with social Mexican jays (*Aphelocoma ultramarina*) out-performing less social Clark's nutcrackers (*Nucifraga columbiana*) [49].

Clark's nutcrackers, considered to be relatively solitary in the wild, are also able to perform a variety of cache protection strategies in the presence of a conspecific [50]. Moreover, other recent evidence suggests that variation in observational spatial memory is more related to a species' dependence on caches than their degree of sociality, as less social but frequent-caching ravens performed above chance levels in an observational spatial memory task, whereas highly social but rarely-caching jackdaws did not [51]. Therefore, the degree to which a corvids' social system influences their socio-cognitive abilities remains unclear. That said, recent research investigating the behavioral flexibility of (highly social) pinyon jays and (less social) Clark's nutcrackers under different social contexts, in which subjects were tested on their caching strategies whilst alone, observed by a conspecific, or observed by a heterospecific, suggests that each species uses different cache protection behaviors. These behaviors seem to be elicited by different social cues, which can be explained in relation to the species' social organization [52]. However, very few studies have explored delayed gratification abilities in a social context, particularly in taxa that differ in sociality.

We aimed to test the flexibility of delayed gratification in a social context in two corvid species - New Caledonian crows (NC crows) and Eurasian jays (E jays) - exploring their choices for immediate vs delayed rewards (varying in quality and preference) while alone compared with in the presence of conspecific(s). We selected these two species as they differ in adult sociality, as outlined above, and they also differ in intensity with which they cache food (NC crows: moderate, i.e. cache variety of food types through-out the year, not entirely dependent on caches for survival; E jays: specialized cachers, i.e. hide large amounts of predominately one food type, usually seasonally available) [24, 53]. Furthermore, both species delay gratification in previous studies, though not tested comparatively with the same paradigm or in a social context. Schnell et al. [54] found that delay of gratification correlated with measures of general intelligence in Eurasian jays. Miller et al. [55] found that New Caledonian crows are better able to delay gratification when rewards varied in quality over quantity and struggled when rewards (immediate or delayed) were not visible compared with being visible.

We used an adapted automatic rotating tray delayed gratification paradigm first introduced in a capuchin (*Cebus apella*) study by Bramlett et al. [56], which we have used previously to test New Caledonian crows and young children by Miller et al. [55], where subjects were required to choose between an immediate reward or wait for a delayed one. The advantage of this paradigm is that it requires minimal pre-training (compared to exchange paradigm) and does not require interaction with an experimenter. While the rotating tray paradigm has not

been used in Eurasian jays previously, this species has been tested using other delay of gratification paradigms (inter-temporal delay maintenance task: Schnell et al., [54]). We used a within-subject, repeated measures design and rewards that differed in quality.

We tested whether corvids can flexibly alter their decision as to whether to wait for a better reward in response to current social conditions, specifically, whilst alone, in a competitive situation (e.g. dominant conspecific), vs a non/less competitive one (subordinate conspecific). We compared behavioural choices between conditions on the individual and species level, and where possible, compared performance between species. Based on our hypothesis that delayed gratification will vary under different social contexts, we predicted that both species may alter their behaviour in the presence of a conspecific compared to being alone, particularly when the conspecific was a competitor. We expected that the birds may wait for the higher-quality (i.e. more preferred) reward when alone (as in Miller et al., [55]) and potentially with a non-competitor conspecific, but may choose the lower-quality, immediate reward (even though less preferred) when a competitor was present (Table 1), as waiting would risk losing the reward to a competitor, leaving the focal bird with nothing. We tested whether there was a difference in performance between species, as their socio-ecological backgrounds (i.e., NCC: more socially tolerant, moderate cachers; EJ: less socially tolerant, specialized cachers) may influence levels of flexibility in delay of gratification across social contexts. However, given the expected increased risk of losing a reward when a competitor was present, we predicted that both species would similarly alter their behaviour in the presence of a competitive conspecific compared to being alone.

## Materials and methods

### Subjects

**New Caledonian crows.** Eleven New Caledonian crows (NC crows) were caught from the wild (at location 21.67˚S 165.68˚E) on Grand Terre, New Caledonia, for temporary holding in captivity on the Island for non-invasive behavioural research purposes from April to August 2019, of which six were available for inclusion in this study. The other five birds were not available as they were engaged in other parallel experiments at the field site, with data collection period limited by season length and experimenter availability. There were three males and three females, based on sexual size dimorphism [57], of which one was adult, two were in their second year (not breeding, remaining in their family group) and three were juveniles (less than 1 year old) (S1 Table). The birds were identifiable with leg-rings (crows were ringed post-capture). During the field season, all crows took part in several experiments, including making forced 2-choices (e.g. between 2 tools or food types) and interacting with artificial apparatuses (e.g. [55]). The birds were housed in a ten-compartment outside aviary, with compartments differing in size, though all at least 2 x 3 x 3m, containing a range of natural enrichment materials like logs, branches and pinecones. Subjects were tested individually in temporary visual isolation from the group, while willingly participating in the study for food rewards to enhance their motivation. The birds were not food deprived and their daily diet consisted of meat, dog food, and fruit, with water available *ad libitum*. The birds maintained at or above capture

Table 1. Predicted selections by condition (social context).

| Condition | Prediction for test trial selection |
| --- | --- |
| Alone (i.e. baseline) | Delayed; higher-quality reward |
| Non/less-competitor | Delayed; higher-quality reward |
| Competitor | Immediate; higher or lower-quality reward |

weights during their stay in captivity. The birds were acclimatized to the aviaries in April and habituated to the experimental apparatus in May, completing the study in August 2019. At the end of their research participation, birds were released at their capture sites. Hunt [58] indicated that New Caledonian crows housed temporarily in a similar situation as the present study successfully reintegrated into the wild after release.

**Eurasian jays.** Eight Eurasian jays (E jays; four males; four females; all adults: S1 Table) participated in this study from September 2022 to May 2023, of which five jays reached criterion for testing. All jays were hand-reared at 10 days old from wild eggs collected by a registered breeder under a Natural England License to NSC (20140062) in 2015. The jays were housed together within a large outdoor aviary (20 m long × 10 m wide × 3 m high) at the Sub-Department of Animal Behaviour, University of Cambridge, Madingley, Cambridgeshire, UK. One end of the aviary was divided into smaller subsections (6 × 2 × 3 m), used to separate mate pairs during the breeding season. Hatch doors connected these subsections to separate indoor testing compartments (each 2 x 1 x 2 m) and could be opened or closed to isolate individuals. Subjects were identified using unique leg-ring color combinations. The jays had *ad libitum* access to water (including during testing) and were fed a mixture of soaked dog or cat biscuits, boiled eggs, boiled vegetables, seeds, and fruit, twice a day. During test days, this food was removed from the aviary approximately 1 hour before testing to increase the birds' motivation to come inside the testing compartments and to participate in experimental trials. The birds were only food restricted for a maximum of 4 hours in one day, although as they habitually cache food, they may have had access to non-test foods during this time. All subjects participated on a voluntary basis (to maximize motivation) and were separated from the group once they entered the testing compartment (by closing the hatch door). When interacting with the birds, the experimenter stood by a window in one of the test compartments.

## Materials

**Apparatus.** The main apparatus used in this experiment was the same as that deployed in Miller et al., [55]. This consisted of a 38 cm diameter raised disk, fitted on top of a rotation device (moving at a speed of 68 s per revolution) which was operated using a remote control (Fig 1). The rotating disk was enclosed within a transparent Perspex box (41 cm × 34 cm × 14 cm) with a rectangular opening at one side (29 cm × 7 cm), to prevent the birds from accessing the rewards until they were positioned directly in front of the subjects. Two small upturned, transparent plastic cups (with a string attached to facilitate cup flipping) covered the rewards and were positioned at two standardized locations on the disk, so that the first reward reached the subject after 5 s (the immediate reward), whereas the second reward reached the subject after 15 s (the delayed reward). Both cups were baited simultaneously. To standardize the position of the birds at the beginning of the trial, the tray was only started once the bird moved to be in front of the tray. The bird made a choice by touching the cup and flipping it to access the reward. Once contact was made with either of the cups, the rotating tray was stopped, meaning they were only allowed to make one choice.

## Procedure

**Pretraining.** *Food preference.* Before the main training stage, the relative preference for each food type was established per individual. To do this, both food types (high-quality: meat, low-quality: apple for NC crows; and high-quality: mealworm, low-quality: bread for E jays) were presented simultaneously in front of each subject (individually isolated in the test compartment). The bird was then allowed to choose one reward and was subsequently prevented from obtaining the other food item. This was repeated for 10 trials per session until the bird

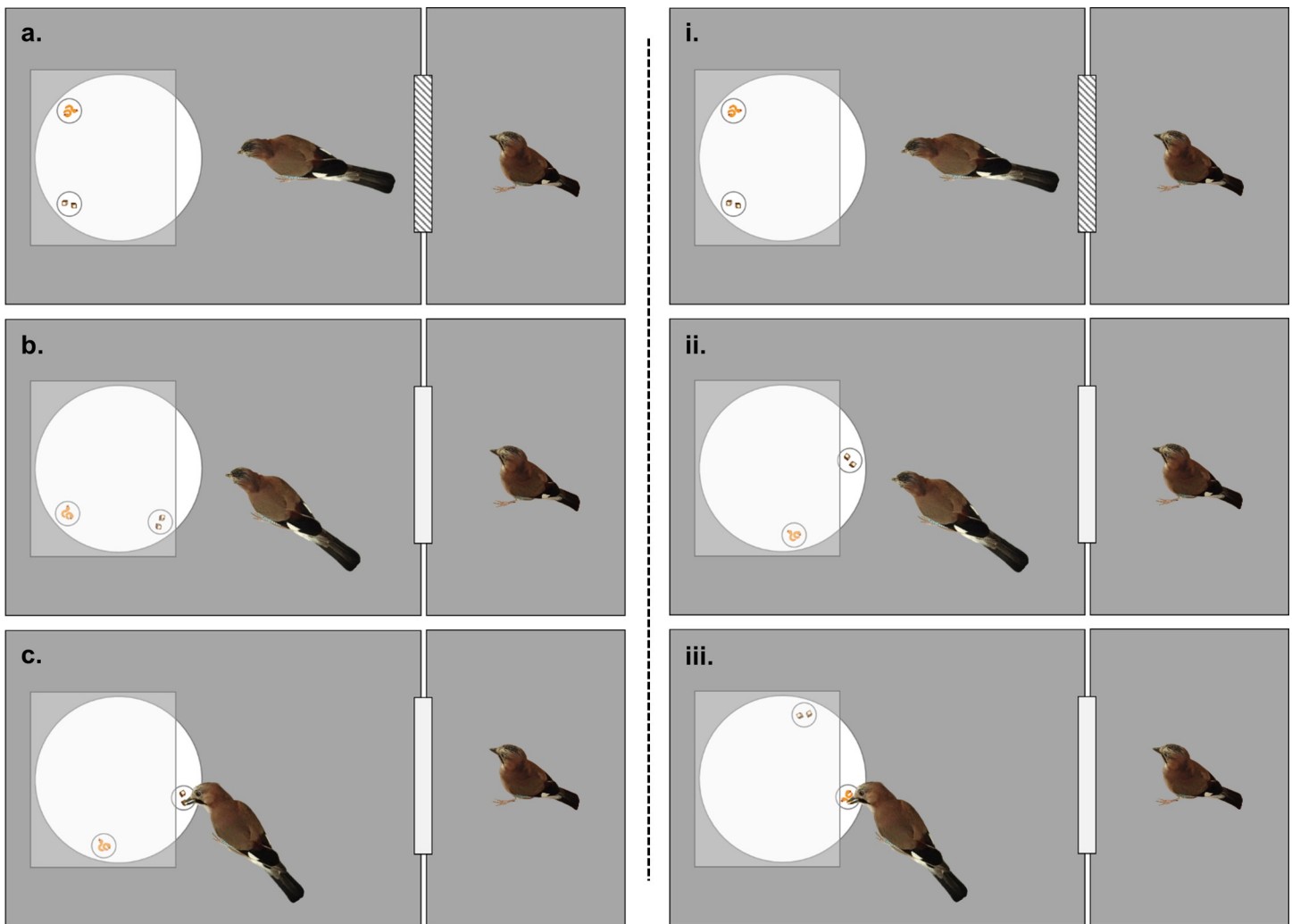

**Fig 1. Diagram representing the potential choices the focal bird could make in test trials. (a-c)**, choosing the immediate option (less preferred choice); **(i-iii)**, choosing the delayed option (more preferred choice). **a) / i)**, Focal bird observes as the rotating tray is baited with both food types (at an equal distance from them) while the competitor observer bird remains in an adjacent compartment with the conjoining door shut. **b)** Just before the first option becomes available, the door between the compartments is opened, allowing the non-focal bird access to the rotating tray. The focal bird then can either choose the immediate option **(c)** or ignore it as it passes **(ii)** and choose the delayed option once it becomes available **(iii)**.

reached the criterion of choosing the high-quality reward 17/20 times (in two consecutive sessions). The position (right or left) of the high-quality reward was pseudorandomized so that it was not in the same location more than twice in a row. We intended to exclude a bird if it did not pass criterion within 10 sessions. However, all six NC crows passed within 2 sessions, and all eight E jays passed within 7 sessions (ranging from 2–7).

*Habituation.* To habituate the birds to the apparatus, they were gradually exposed to the apparatus in multiple phases; progressing each phase when they began taking food comfortably. First, the tray remained turned off (and so not moving) with the food placed near it. Then, the apparatus was switched on (moving) with food again placed near it. Next, the food was placed on top of the moving tray. Finally, the food was placed on top of the moving tray and the experimenter turned the tray off and on again (after each piece of food was collected) to habituate the birds to the sound the tray makes when stopping and to tray movement. Each

phase was done as a group (with each individual free to leave the compartment) and then subsequently as an individual (separated from the group within the compartment).

*Forced choice training.* For the birds to learn that they could only make one choice of food (causing the tray to stop) in each trial, they were given trials in which only one cup was baited and the other remained empty. As such, if the food was in the delayed position, and the bird selected the immediate cup, then they did not receive a reward. In one session of 10 trials, the rewarded cup was placed at the immediate location 5 times and at the delayed location 5 times, in a pseudorandomized order (so that the reward was not in the same location more than twice consecutively). The birds passed criterion for this phase when they chose the food in the delayed position in 9/10 trials across two consecutive sessions. If they failed to pass this criterion within 15 sessions (i.e., 150 trials) then they were discounted from the experiment. However, all six NC crows passed within 2 sessions, and all eight E jays passed within 14 sessions (ranging from 6–14).

*Food monopolization.* Before being tested in the test conditions, food monopolization tests were conducted to assess the relative dominance relationships between pairs of individuals to inform the assignment of non-focal birds (competitor/non-competitor) in these trials (S1 and S2 Tables). This was always done between two individuals isolated from the rest of the group. Choices of which birds to test as non-focal birds were informed by general observations of displacement and other competitive behaviors under non-test conditions. As we tested relative dominance, a single individual could be both a competitor and a non-competitor observer depending on the identity of the focal bird that they were paired with. To confirm the dominance ranking within the pair in food monopolization trials, the experimenter baited a cup on a platform whilst both birds observed, then simultaneously allowed both birds access to the baited cup. If the focal bird took the food without being displaced, then the non-focal bird was considered to be a non-competitor, but if the focal bird was displaced or did not attempt to obtain the food, then the non-focal bird was considered to be a competitor. Food monopolization trials were sometimes repeated (for the jays) immediately before test trials if observations suggested that the dominance relationships may have changed and non-focal birds re/assigned accordingly.

**Testing.**  Upon successful completion of the forced choice phase and food monopolization trials, the birds began the test phase. This phase was made up of trials in three different conditions: 'alone', 'non-competitor', and 'competitor'. Each bird received 2 sessions per test condition (totaling to 20 trials each). In each session, 8/10 trials were 'test' trials (in which the high-quality reward was in the delayed position, and the low-quality reward was in the immediate position) and the remaining 2/10 trials were 'control' trials (in which the high-quality reward was in the immediate position, and the low-quality reward was in the delayed position). Each individual received both alone sessions first, then the remaining two social conditions. The order in which the birds received the non-competitor and competitor sessions was counterbalanced across individuals. The stimulus birds were selected opportunistically and in accordance with the relationships determined by the food monopolization tests, and so occasionally varied between replicates (note that what is important here is not the identity of the stimulus bird, but their relationship with the focal bird). The conditions were then alternated every session for each bird (e.g., non-competitor, competitor, non-competitor, competitor). A choice was made once the bird touched either cup and were recorded as an immediate choice (Fig 1A–1C and S1A File), a delayed choice (i-iii; S1B File), or no choice (as the non-focal bird took either reward before the focal bird could or displaced the focal bird; no choices = competitor trials: n = 19, non-competitor trials: n = 1; 4c in S1 File). During the social conditions, while the rotating tray was baited with food rewards, the competitor/ non-competitor observer bird remained in an adjacent compartment with the conjoining door shut. Before the immediate

option became available to the focal bird, the observer bird was also allowed access to the rotating tray, and the focal bird's choice (immediate or delayed reward) was recorded. A timeline of the pretraining and testing phases can be found in S1 Fig. By design, there was a minimum and maximum of two sessions each for the competitor and non-competitor conditions.

*Alone*. The birds first received alone trials to assess their baseline ability to delay gratification in a non-social context, as in these trials the bird was alone in the testing compartments. The six NC crows selected the high-quality reward in the 13/16 test choices within 2 sessions (S3 Table). However, the E jays required additional training to successfully complete these baseline trials and therefore E jays' sessions were repeated until an individual made 13/16 test choices (high-quality reward was at the delayed position) to the delayed reward in two consecutive sessions. These last two sessions were then used as the alone test condition. However, if the E jays did not reach this criterion in 15 sessions, they were excluded from the experiment. Five (three females; two males) of eight jays met this criterion (ranging from 3–8 sessions). We calculated 'learning speed' based on the number of trials to reach criterion in the alone condition (S3 Table).

*Non-competitor*. In these trials, the focal birds were tested with a non-competitor conspecific (determined by the food monopolization trials–see earlier) in an adjacent compartment. The non-focal bird was allowed access to the main test compartment (with the apparatus) just before the immediate reward became accessible (Fig 1). A trial was terminated once the focal bird made a choice.

*Competitor*. In these trials, the focal birds were tested with a conspecific competitor (determined by the food monopolization trials) in an adjacent compartment. The non-focal bird was allowed access to the main test compartment (with the apparatus) just before the immediate reward became accessible (Fig 1). A trial was terminated once the focal bird made a choice or was displaced by the non-focal bird (no choice).

## Data analysis

We recorded the choice per trial for each subject as 'immediate' (1) or 'no choice/ delayed' (0), with proportion over total number of trials (control and test trials). All test sessions were coded live as well as being video recorded. Example trials can be found in S1 File.

We conducted Linear Mixed Models (LMM: [59] with binomial distribution using R (version 2023.03.0+386, [60]) to assess which factors influenced choices in the New Caledonian crows and Eurasian jays. Choice was a binary variable indicating whether the subject selected immediate (1) or delayed/ no reward (0) per trial and was entered as a dependent variable in the model. For the model, we included the random effect of subject ID and fixed effects of species (NC crows, E jays), condition (alone, competitor, non-competitor), with interaction effects of species*condition. We used the test trial data (high-quality reward in delayed position; low-quality reward in immediate position). In control trials, all subjects selected the immediate, high-quality reward irrespective of condition (100% of trials). We used Tukey comparisons for post-hoc comparisons (package multcomp, function dlht ()) and the DHARMa package [61] to test model assumptions. The model did not fail to converge, with a confidence interval of 97.5%. Model assumption checks showed no deviation from expected distribution. For individual-level analysis, we used exact two-tailed Binomial tests of choices (delayed) per condition (SPSS version 28).

## Ethics statement

The study methods were conducted in accordance with relevant guidelines and regulations. The Eurasian jay study was reviewed and approved by the University of Cambridge Animal

Welfare Ethical Review Body (AWERB) and was conducted under a non-regulated license (NR2022/82). The New Caledonian crow research was conducted under approval from the University of Auckland Animal Ethics Committee (reference number 001823) and from the Province Sud with permission to work on Grande Terre, New Caledonia, and to capture and release crows.

## Results

### Group-level performance: Testing effects of condition and species

At the group level, selection of the low-quality, immediate option differed between species (LLM: $\chi2$ = 168.75, d.f = 2, p <0.0001), by condition ($\chi2$ = 52.49, d.f = 1, p <0.0001) and within condition by species interaction ($\chi2$ = 60.36, d.f = 2, p <0.0001). The jays were more likely to select the low-quality, immediate reward than the crows (Tukey contrasts: E jays - NC crows, z = 2.66, p = 0.00782). The jays were also more likely to select the low-quality, immediate reward when they were with a non-competitor than when alone (z = 2.676, p = 0.00745), but the difference was stronger when a competitor was present than when they were alone (Tukey contrasts: z = 7.270, p <0.0001), as well as with a competitor than a non-competitor (z = -4.616, p < 0.0001: Fig 2). The crows were not more likely to select the low-quality, immediate reward depending on condition, i.e. they selected the delayed, high-quality reward irrespective of condition (Tukey contrasts: alone - competitor, z = -0.196, p = 0.845; alone - noncompetitor, z = -1.040, p = 0.298; noncompetitor vs competitor, z = -0.864, p = 0.388).

### Individual-level performance: Selection of high-quality, delayed reward by condition

On an individual level, all six NC crows selected the high-quality, delayed reward over the low-quality, immediate reward in all three conditions (Table 2). In contrast, while all five E jays selected the high-quality, delayed reward while alone, no jays significantly chose the delayed reward while a competitor or non-competitor was present. Rather, the E jays changed their behaviour by selecting the low-quality, immediate reward in some trials (Table 2). One jay (Stuka) switched strategy entirely when a competitor was present - significantly selected the immediate over the delayed reward.

## Discussion

We tested the flexibility of the ability to employ delayed gratification, i.e. to wait for a delayed, higher-quality reward over an immediate, lower-quality one, in different social conditions in two corvid species that differ in sociality and food-caching, New Caledonian crows and Eurasian jays, using the rotating-tray paradigm. We found species and condition differences on choices to select an immediate, but lower-quality reward over a delayed, higher-quality one. Specifically, jays were more likely to select the immediate, low-quality reward than crows. Jays, though not crows, were also more likely to alter their choices while alone compared with when a competitor or a non-competitor was present. Crows continued to forgo the immediate, lower-quality reward for the delayed, higher-quality one irrespective of condition. Our findings highlight that the ability to delay gratification in Eurasian jays is influenced by the presence of conspecifics, depending on their identity (competitor/ non-competitor), suggesting flexibility in their delayed gratification abilities. On the other hand, the crows continue to delay gratification even with a competitor present, reflecting stability (or inflexibility) in their delayed gratification abilities. Furthermore, both species were capable of delaying gratification

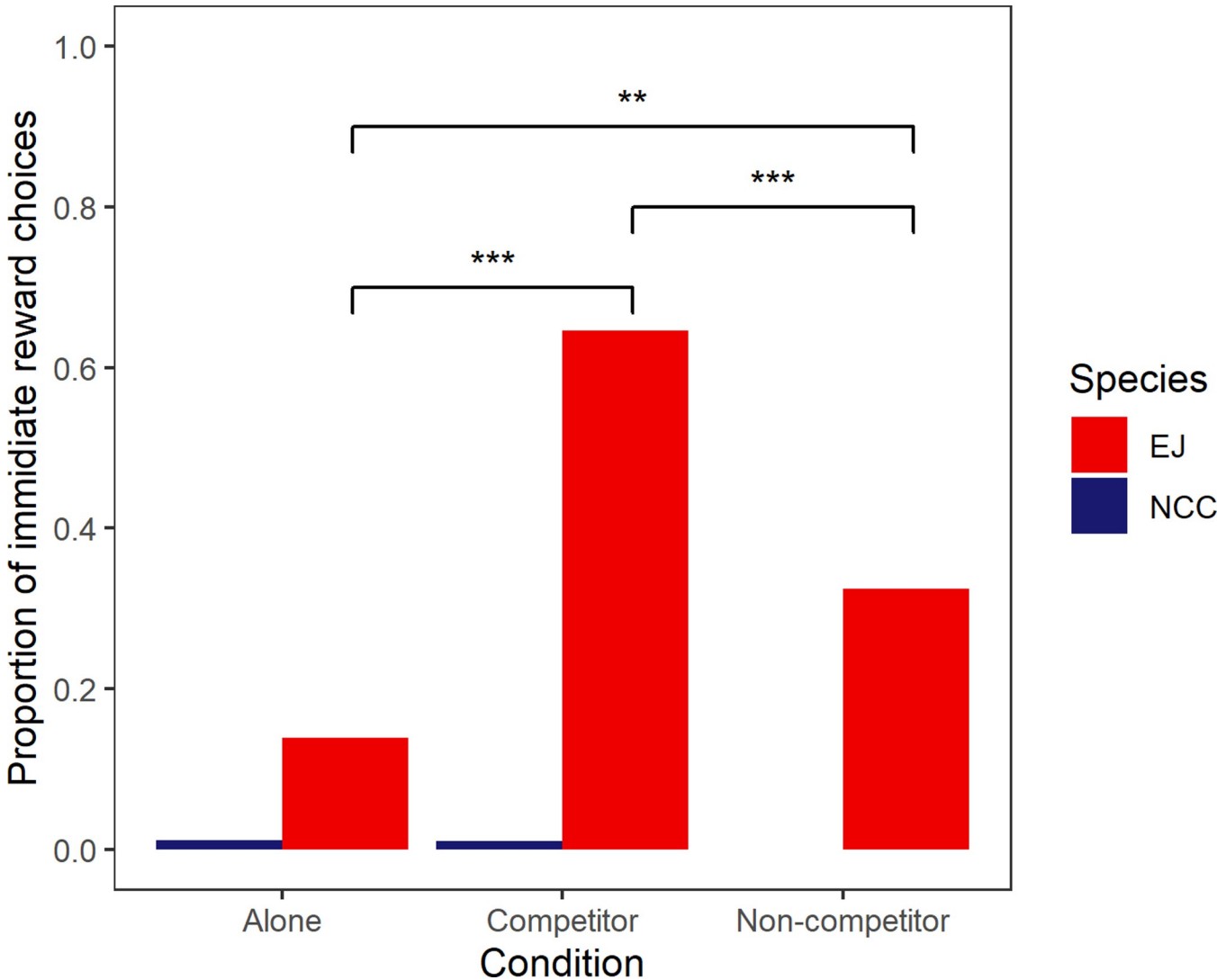

**Fig 2. Proportion of choices of the immediate (low-quality) reward per condition for Eurasian jays (EJ) and New Caledonian crows (NCC).** ** $p > 0.01$; *** $p > 0.001$.

in this paradigm, comparable with young children and other New Caledonian crows in a previous study [55] (S2 File), as well as capuchin monkeys (*Cebus Apella*) [56].

The species difference was unexpected, with the crows selecting the delayed, high-quality reward regardless of social condition, while the jays altered their choices when competitors or non-competitors were present. Both species were able to reliably delay gratification while alone, which was expected, given New Caledonian crows delayed gratification using the rotating tray paradigm in a previous study [55]. We note the jays took longer to train than the crows (crows: 2 sessions; jays 3–8 sessions to pass criterion) and three other jays did not pass criterion to proceed to testing (despite having 15–34 sessions of 10-trials per session). It is possible that training length was influenced by neophobia differences between species, as the jays are typically more neophobic than the crows [53]. Future research may expand on samples and data set size to explore potential differences in species' learning speed.

**Table 2. Delayed choices per individual across conditions for test trials only (high-quality reward in delayed position).**

| ID | Species | Choice | Alone (out of 16) | Competitor (out of 16) | Non-competitor (out of 16) | % Overall |
|---|---|---|---|---|---|---|
| Birute | NC crows | Delayed | 16 | 16 | 16 | 100 |
| | | p-value | <**0.0001** | <**0.0001** | <**0.0001** | |
| Fossey | NC crows | Delayed | 16 | 16 | 16 | 100 |
| | | p-value | <**0.0001** | <**0.0001** | <**0.0001** | |
| Irene | NC crows | Delayed | 16 | 16 | 16 | 100 |
| | | p-value | <**0.0001** | <**0.0001** | <**0.0001** | |
| Konrad | NC crows | Delayed | 16 | 16 | 16 | 100 |
| | | p-value | <**0.0001** | <**0.0001** | <**0.0001** | |
| Leakey | NC crows | Delayed | 16 | 16 | 16 | 100 |
| | | p-value | <**0.0001** | <**0.0001** | <**0.0001** | |
| Marie | NC crows | Delayed | 15 | 15 | 16 | 95.83 |
| | | p-value | **0.0005** | **0.0005** | <**0.0001** | |
| Godot | E jays | Delayed | 13 | 7 | 12 | 68.75 |
| | | p-value | **0.0213** | 0.804 | 0.0768 | |
| Homer | E jays | Delayed | 15 | 5 | 12 | 68.75 |
| | | p-value | **0.0005** | 0.210 | 0.0768 | |
| Penny | E jays | Delayed | 13 | 9 | 9 | 64.58 |
| | | p-value | **0.0213** | 0.804 | 0.804 | |
| Sojka | E jays | Delayed | 14 | 5 | 11 | 62.5 |
| | | p-value | **0.0042** | 0.210 | 0.2101 | |
| Stuka | E jays | Delayed | 13 | 2 | 10 | 52.08 |
| | | p-value | **0.0213** | *0.004* | 0.455 | |

Binomial exact two-tailed test: p = <0.05 highlighted in bold. NC crows = New Caledonian crow; E jays = Eurasian jay. In one case, Stuka made a majority of immediate choices highlighted in italics as significant immediate, low-quality reward choice.

It is also possible that species differences were related to limitations of the study set-up or subject sourcing. Although both species were originally sourced in the wild, the jays had been hand-reared and housed long-term in captivity, whereas the crows were parent-reared and only temporarily held in captivity ($\sim$4 months). Both species received adequate habituation and were required to pass comparable criterion prior to testing. Furthermore, although the crows were more recently sourced from the wild, they were tested and showed high performances in several other cognitive experiments during this field-season (e.g. [55]), indicating they were well habituated for testing before participating in the current study. The jays were all adults, while the crows ranged in age (juvenile to adult). Whilst we are not aware of any studies investigating the development of delay of gratification in corvids, an experiment with human children, using the same rotating-tray task, shows evidence for age-related improvements in delay of gratification ability across cultures [62]. Therefore, development may also play a role in the birds' performance in this task. However, there were no differences in choices between individual crows (Table 2) and we do not have sufficient variation in the jay performance to test for age effects.

The type of competitor/ non-competitor was as comparable as possible between species. All subjects were familiar with their observing, non-focal conspecifics (NC crows caught together so potentially a family unit) although the prior interactions of the NC crows were unknown (being wild caught) (S1 Table). However, in the jays, the non-focal/ observer bird (competitor/ non-competitor) was not always the same individual across all trials, partly due to practical issues of encouraging the focal and non-focal to participate in each trial and partly due to

more fluid dominance relationships. With the jays, it appeared that the dominance relationships varied between some pairs across the 6-month period of this study, hence, we conducted repeated food monopolization trials and assigned the non-focal accordingly (S1 and S2 Tables). We also note that many of the jay breeding pairs in this captive colony change year-by-year. For both species, the food monopolization trials supported the distinction of a competitor versus non-competitor status for the non-focal bird in relation to the focal bird. The food monopolization trials for the crows were limited to the group and conducted prior to testing due to field season time pressures. The crow test compartments were around twice the size of the jay compartments, so it is possible that the crow non-focal took longer to reach the platform, thereby potentially less likely to directly compete for rewards. The focal and non-focal (both species) were released simultaneously though to remedy this issue. Furthermore, we incorporated the requirement for the focal to lift a small lid to obtain the reward, once chosen, which created a short time delay between selection and eating/hiding the reward in their bill.

The species differed from one another in adult sociality (NC crows: family groups; E jays: territorial pairs) and food caching (NC crows: moderate; E jays: specialised cachers) [24, 53]. We selected adult sociality as it is more consistent than at the juvenile/ subadult stages and our sample consists primarily of adults. These socio-ecological factors could impact choices relating to food selection and responses to competition. We interpret these findings as a caching specialist with territorial pair living (E jays) showing flexibility or perhaps struggling to delay gratification when there is social competition, while a moderate caching and family-group living species (NC crows) continues to delay gratification - suggesting stability (or inflexibility) in behaviour regardless of social context. This flexibility by the E jays may relate to this change in behaviour being a more adaptive response to take any reward available immediately (even if less preferred), rather than risk waiting and end up without any reward at all, as the competitor may take it.

With regard to caching, the jays - being specialised cachers - have evolved under the social context of cache pilfering and development of cache protection strategies [46]. With sociality, the jays may be less tolerant of potential competitors, being more likely to actively displace conspecifics and defend territories, than the crows. Although not a highly social corvid species, the New Caledonian crows may form temporary aggregations of small groups [63] and will tolerate conspecifics outside of their family groups - largely juveniles and sub-adults (2-years old) - with rarely observed aggressive interactions [64]. Juvenile crows have been observed showing submissive displays when in the presence of non-family adults [64]. It is possible that stable hierarchies exist with the crows [64], similar to carrion crows (*Corvus corone*) [65]. This is less likely with the Eurasian jays, given the variation observed in the food monopolization trials and continuous changing of breeding pairs suggesting non-linear hierarchies (S2 Table), as well as the generally dyadic and territorial nature of Eurasian jays in the wild [24]. Species differences in responses to novel food and objects (i.e. neophobia) may influence testing performance [53], however, this is unlikely due to habituation and both species demonstrating a reliable ability to delay gratification in the alone condition (Table 2).

The condition effect in the Eurasian jays was largely in line with our expectations. The jays flexibly altered their choices depending on the social context, being more likely to take the immediate reward, even though of lower quality, rather than risk losing it to a competitor. They were more strongly influenced by a competitor than a non-competitor on the group-level. However, on the individual level, all five jays did not show significant differences between competitor and non-competitor trials as they still chose the immediate, low-quality reward in some trials in both conditions (Table 2). These findings may relate to a higher risk of being displaced and losing the reward to any conspecific.

These captive jays were hand-reared socially and live most of the time (outside of breeding season, when they live in pairs to reduce risk of aggression) in a large social group. This social setting is quite different to their natural behaviour in the wild, where when adult, they will largely defend territories in pairs [24]. Furthermore, in captivity, they are provided with adequate food for all individuals, distributed through-out the large aviary to reduce any competition. Whether or not jays living in the wild would also show this flexibility in behaviour in response to social context requires future focus. Regardless of these aspects of the captive setting, the jays appear to pay attention and respond to the presence and identity of others while delaying gratification, while the crows do not adjust their choices according to social competition. These findings are in line with previous studies on Eurasian jays testing flexibility of other behaviours in social contexts. For example, they are able to switch caching and pilfering behaviour depending on whether they are more subordinate or dominant than a conspecific present [46]. In addition, evidence suggests that Eurasian jays are also capable of desire state attribution towards both their partners and competitors [36, 66, 67] (although see [47]).

Future research can expand on species comparisons to explore social influences on self-control and other aspects of decision-making. For instance, using the rotating tray paradigm or other delayed gratification paradigms in non-human primates and human children, or in highly social/tolerant species compared with less social/ tolerant ones within taxa. Expanding on the length of delay, as this study utilised only a short delay (15 seconds), the quantity (as we only tested using quality differences) and visibility of rewards provides several avenues for future work. Furthermore, it would be worthwhile to expand on the identity of the observer, for instance, to see whether familiarity or age influences choices in delayed gratification tests, in particular, whether NC crow delayed gratification is influenced by presence of other types of observers.

## Conclusion

In conclusion, we explored the effect of social influences on delayed gratification in two corvid species - New Caledonian crows and Eurasian jays - highlighting both species and condition (alone, competitor, non-competitor) differences in performance. Both species were able to delay gratification. The jays did so flexibly depending on the social context, while the crows remained stable in their choices for delayed rewards. These findings contribute to our understanding of self-control and the factors influencing delayed gratification in non-human animals. In particular, flexibility (or inflexibility) in delay of gratification varies under differing social contexts, which may relate to the species' social tolerance and related risk of competition.

## Supporting information

**S1 Table. Subject information.** *Change in relative dominance between sessions (see S2 Table).
(DOCX)

**S2 Table. Food monopolization results for the Eurasian jays.** *Change in relative dominance between two specific individuals.
(DOCX)

**S3 Table. 'Learning speed' per individual and species: Number of trials and sessions to reach criterion and complete test trials (last 2 sessions counted) in alone condition.** * = did not reach criterion: three of eight jays did not reach criterion within 15 sessions (*) so were excluded from further testing. NC crows = New Caledonian crows; E jays = Eurasian jays.
(DOCX)

**S1 Fig. Timeline of pretraining and testing phases.** Coloured arrows show the different test condition sequences that individuals were assigned to for sessions one through to four (S1-S4). The dotted line shows a possible repeat of the food monopolization phase if relative dominance relationships were perceived to change mid-test sequence (jays only).
(DOCX)

**S1 File. Example video trials for both species.**
(DOCX)

**S2 File. Comparison of baseline to Miller et al. 2020 [55].**
(DOCX)

## Acknowledgments

Thank you to Ian Millar for help in apparatus construction and to Alizée Vernouillet for assisting in initial related training of Eurasian jays. Thanks to Province Sud for the permission to work in New Caledonia, and to Dean M. and Boris C. for granting property access for catching and releasing the crows.

## Author Contributions

**Conceptualization:** Rachael Miller.

**Data curation:** Rachael Miller, James R. Davies.

**Formal analysis:** Rachael Miller, James R. Davies.

**Funding acquisition:** Rachael Miller, Alex H. Taylor, Nicola S. Clayton.

**Investigation:** Rachael Miller, James R. Davies, Martina Schiestl, Elias Garcia-Pelegrin.

**Methodology:** Rachael Miller.

**Project administration:** Rachael Miller, Nicola S. Clayton.

**Resources:** Rachael Miller, Russell D. Gray, Alex H. Taylor, Nicola S. Clayton.

**Supervision:** Rachael Miller, Alex H. Taylor, Nicola S. Clayton.

**Validation:** Rachael Miller, James R. Davies.

**Visualization:** Rachael Miller, James R. Davies.

**Writing – original draft:** Rachael Miller, James R. Davies.

**Writing – review & editing:** Rachael Miller, James R. Davies, Martina Schiestl, Elias Garcia-Pelegrin, Russell D. Gray, Alex H. Taylor, Nicola S. Clayton.

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
