## [Decision Letter · Decision Letter 0]

21 Sep 2023

PONE-D-23-21809Social influences on delayed gratification in New Caledonian crows and Eurasian jaysPLOS ONE

Dear Dr. Miller (Harrison),

Thank you for submitting your manuscript to PLOS ONE. After careful consideration, we feel that it has merit but does not fully meet PLOS ONE’s publication criteria as it currently stands. Therefore, we invite you to submit a revised version of the manuscript that addresses the points raised by the reviewer.  I apologize for the delay and review process.  My second reviewer was unable to complete your review, and unfortunately did not notify me quickly.  I have read and evaluated your manuscript myself, and I concur with reviewer #1 suggestions. I would only add that you explain "effectively comparatively" in Lines 126-129: where you write, "The advantage of this paradigm is that it requires minimal pre-training (compared to exchange paradigm), does not require interaction with an experimenter and can be run effectively comparatively.”  Last, I suggest you discuss the role of development in delayed gratification, as your birds are of mixed ages.

We look forward to receiving your revised manuscript.

Kind regards,

Brenton G. Cooper, Ph.D.

Academic Editor

PLOS ONE

4. We note that Figure 1 in your submission contain copyrighted images. All PLOS content is published under the Creative Commons Attribution License (CC BY 4.0), which means that the manuscript, images, and Supporting Information files will be freely available online, and any third party is permitted to access, download, copy, distribute, and use these materials in any way, even commercially, with proper attribution. For more information, see our copyright guidelines: http://journals.plos.org/plosone/s/licenses-and-copyright.

Reviewers' comments:

Reviewer's Responses to Questions

**Comments to the Author**

1. Is the manuscript technically sound, and do the data support the conclusions?

Reviewer #1: Partly

2. Has the statistical analysis been performed appropriately and rigorously? 

Reviewer #1: Yes

3. Have the authors made all data underlying the findings in their manuscript fully available?

Reviewer #1: No

4. Is the manuscript presented in an intelligible fashion and written in standard English?

Reviewer #1: Yes

5. Review Comments to the Author

Reviewer #1: This paper considers an interesting topic of social influences on delayed gratification for a food incentive. Two corvid species were considered, each with different social and food caching behaviours. The study did find species differences in delayed gratification, depending on the context (treatment) but the outcomes might also reflect other confounding factors.

Points to consider.

Abstract. Provide the hypotheses (or predictions) tested.

L 39-40. Provide a summary of the key conclusions. The abstract should be self-contained and not refer to the details later.

Introduction

L 65-66. Split into 2 sentences.

L 92, 100. Don't start a new paragraph with a joining word.

L 97. Steal is anthropomorphic

L 109. Change to: each species uses.

L 118. Delete: have been found to be able to

Provide a prediction (or a hypothesis) for species level differences in delayed gratification - related to levels of sociality/caching? Without this, there is no basis statistical comparisons between species.

Materials

L 159. I assumed that they were ringed post-capture.

L 186-189. What about the crows? Were they also subjected to a similar training process?

Methods

L 209 onwards. It is really difficult to follow the timing and duration of the training and testing phases. A timeline diagram will be useful.

L 226. Since none of the birds were excluded, change the wording to: We intended to exclude a bird if it did not pass the criterion within 10 sessions. However, ...

L 253. Did you establish a dominance hierarchy or simply dominance relationships?

L 264. Make clear whether the same stimulus birds were used in each replicate.

L 274. Mention this part about the experimental set-up in the methods too.

L 282. Change to: and therefore ...

L 289, 294. In these sections, provide the range of times taken to complete these treatments (also for the alone treatment).

Results

It was a good idea to provide group and individual level differences.

Discussion

The discussion is focussed on the limitations of the species origin, housing and training differences. This is understandable given the many confounding effects. The part that considers species differences on which the study is premised (but not predicted) is sociality and caching and is discussed briefly in L 401 to 406. The challenge for the authors is to separate experimental conditions from underlying species effects. The manuscript does not quite handle this adequately. The NC did not show any obvious social influences on delayed gratification but is this due to their natural behaviour or their recent capture and possibly previous familiarity with other birds (as already alluded to in the discussion). The jays took longer to learn. Some issues to address: does the time to learn predict the outcome in the various treatments. Can this be tested in the species - perhaps using a learning model . I realise that the sample size is small, but I do suggest that you consider assessing the effects of some of the confounding variables, even if descriptively.

Conclusion

L 456. In the last sentence, I suggest adding ... in non-human animals.

6. PLOS authors have the option to publish the peer review history of their article (what does this mean?). If published, this will include your full peer review and any attached files.

Reviewer #1: No

---

## [Author Response · Author response to Decision Letter 0]

9 Oct 2023

EDITOR COMMENTS:

Done. 

Done. 

Done.

4. We note that Figure 1 in your submission contain copyrighted images. All PLOS content is published under the Creative Commons Attribution License (CC BY 4.0), which means that the manuscript, images, and Supporting Information files will be freely available online, and any third party is permitted to access, download, copy, distribute, and use these materials in any way, even commercially, with proper attribution. For more information, see our copyright guidelines: http://journals.plos.org/plosone/s/licenses-and-copyright.

Figure 1 was created entirely by James Davies (cc’d) who is a named author on this paper - he made the images out of shapes (including the mealworms) and took the photos of the jays himself. He has the original designs and images if needed. 

We also double checked with the PLOS ONE team who replied: “Thank you for reaching out! We apologize for the confusion. If Figure 1, or any figure, was created entirely by a study author, you should just be sure to include that information in your Response to Reviewers document. Any potential copyright issues will be investigated by journal staff upon resubmission, and if all elements of an image were created by a study author, there will be no issue. Just be sure to explain the origin of the image in your Response to Reviewers.”

Done. 

EDITOR COMMENTS CONTINUED:

I would only add that you explain "effectively comparatively" in Lines 126-129: where you write, "The advantage of this paradigm is that it requires minimal pre-training (compared to exchange paradigm), does not require interaction with an experimenter and can be run effectively comparatively.” 

Edited to remove “can be run effectively comparatively”, as the main advantages of this paradigm are already outlined. 

Last, I suggest you discuss the role of development in delayed gratification, as your birds are of mixed ages.

Done – added as: “Whilst we are not aware of any studies investigating the development of delay of gratification in corvids, an experiment with human children, using the same rotating-tray task, shows evidence for age-related improvements in delay of gratification ability across cultures [62]. Therefore, development may also play a role in the birds’ performance in this task. However, there were no differences in choices between individual crows (Table 2), and we do not have sufficient variation in the jay performance to test for age effects” (LINE 467-473)

Reviewers' comments:

Reviewer's Responses to Questions

Comments to the Author

1. Is the manuscript technically sound, and do the data support the conclusions?

Reviewer #1: Partly

2. Has the statistical analysis been performed appropriately and rigorously? 

Reviewer #1: Yes

3. Have the authors made all data underlying the findings in their manuscript fully available?

Reviewer #1: No

Please note that we did provide access to the full data set and R script under the data availability section: i.e. “The full data set and R script is available on Figshare: 10.6084/m9.figshare.23514828 (private link: https://figshare.com/s/3a6adfae2cb31f707659).” If there is any access issue, please do let us know so we can rectify this further. 

4. Is the manuscript presented in an intelligible fashion and written in standard English?

Reviewer #1: Yes

5. Review Comments to the Author

Reviewer #1: This paper considers an interesting topic of social influences on delayed gratification for a food incentive. Two corvid species were considered, each with different social and food caching behaviours. The study did find species differences in delayed gratification, depending on the context (treatment) but the outcomes might also reflect other confounding factors.

Thank you for your helpful and constructive feedback. 

Points to consider.

Abstract. Provide the hypotheses (or predictions) tested.

Done – added to abstract as: “We predicted that, given the increased risk of losing a reward with a competitor present, both species would similarly, flexibly alter their choices in the presence of a conspecific compared to when alone.”. 

L 39-40. Provide a summary of the key conclusions. The abstract should be self-contained and not refer to the details later.

Done – added to abstract as: “New Caledonian crows are more socially tolerant and moderate cachers, while Eurasian jays are highly territorial, specialized cachers. Both species have evolved under the social context of cache pilfering and cache protection strategies. Therefore, the flexibility (or inflexibility) in delay of gratification under different social contexts may relate to the species’ social tolerance and related risk of competition.”

Introduction

L 65-66. Split into 2 sentences.

Done.

L 92, 100. Don't start a new paragraph with a joining word.

Done.

L 97. Steal is anthropomorphic

Changed to “take”. 

L 109. Change to: each species uses.

Done.

L 118. Delete: have been found to be able to

Done.

Provide a prediction (or a hypothesis) for species level differences in delayed gratification - related to levels of sociality/caching? Without this, there is no basis statistical comparisons between species.

Done – we have added the following species prediction: “We tested whether there was a difference in performance between species, as their socio-ecological backgrounds (i.e., NCC: more socially tolerant, moderate cachers; EJ: less socially tolerant, specialized cachers) may influence levels of flexibility in delay of gratification across social contexts. However, given the expected increased risk of losing a reward when a competitor was present, we predicted that both species would similarly alter their behaviour in the presence of a competitive conspecific compared to being alone.” (LINES 177-184)

Materials

L 159. I assumed that they were ringed post-capture.

Added in. 

L 186-189. What about the crows? Were they also subjected to a similar training process?

Edited from “jays” to “birds”

Methods

L 209 onwards. It is really difficult to follow the timing and duration of the training and testing phases. A timeline diagram will be useful.

Great idea, thanks. We’ve now added a timeline diagram to the supplementary materials (S3 Figure). 

L 226. Since none of the birds were excluded, change the wording to: We intended to exclude a bird if it did not pass the criterion within 10 sessions. However, ...

Done

L 253. Did you establish a dominance hierarchy or simply dominance relationships?

We established the dominance relationships between pairs of individuals – we clarify this in the text and remove reference to “hierarchy” to avoid confusion. 

L 264. Make clear whether the same stimulus birds were used in each replicate.

We have now added this sentence to the “testing” paragraph as suggested: “The stimulus birds were selected opportunistically and in accordance with the relationships determined by the food monopolization tests, and so occasionally varied between replicates (note that what is important here is not the identity of the stimulus bird, but their relationship with the focal bird).”

L 274. Mention this part about the experimental set-up in the methods too.

Done. 

L 282. Change to: and therefore ...

Done. 

L 289, 294. In these sections, provide the range of times taken to complete these treatments (also for the alone treatment).

If ‘the range of times’ is referring to the number of sessions taken to complete the treatment, then this information is provided for the alone condition (LINES 340-348) and, by design, is a minimum and maximum of two sessions each for the competitor and non-competitor conditions (added to LINES 325-326). 

If ‘the range of times’ is referring to the number of days/weeks taken to complete the treatment, then we do not think that this information is particularly relevant, as it does not relate to the hypotheses we are testing and varied due to practical details. This is because, for the jays, we tested them on a voluntary and opportunistic basis, i.e., when they decided to enter the test rooms. For this reason, some individuals did not participate in trials for periods of time. Therefore, timing information does not relate to learning trajectories or performance. The range of days it took to complete the conditions is comparable between species and periods of absence were mixed across test conditions. We provide the number of trials and sessions already in the supplementary materials. 

Results

It was a good idea to provide group and individual level differences.

Thank you. 

Discussion

The discussion is focussed on the limitations of the species origin, housing and training differences. This is understandable given the many confounding effects. The part that considers species differences on which the study is premised (but not predicted) is sociality and caching and is discussed briefly in L 401 to 406. The challenge for the authors is to separate experimental conditions from underlying species effects. The manuscript does not quite handle this adequately. The NC did not show any obvious social influences on delayed gratification but is this due to their natural behaviour or their recent capture and possibly previous familiarity with other birds (as already alluded to in the discussion). The jays took longer to learn. 

We now add specific hypothesis and predictions relating to species, as you suggested, as outlined in the above responses to comments, e.g. in the abstract and intro. We also address the learning/ training point below. Regarding the point re: study limitations vs wider socio-ecological factors, we also added to the discussion: “Furthermore, though the crows were more recently sourced from the wild, they were tested and showed high performances in several other cognitive experiments during this field-season (e.g. [55]), indicating they were well habituated for testing before participating in the current study.” (LINES 464-466)

Some issues to address: does the time to learn predict the outcome in the various treatments. Can this be tested in the species - perhaps using a learning model . I realise that the sample size is small, but I do suggest that you consider assessing the effects of some of the confounding variables, even if descriptively.

We had addressed this point in the discussion: “It is possible that training length was influenced by neophobia differences between species, as the jays are typically more neophobic than the crows [53]. Future research may expand on samples and data set size to explore potential differences in species’ learning speed.” (LINES 457-459)

Conclusion

L 456. In the last sentence, I suggest adding ... in non-human animals.

Done.

---

## [Editor Report · Decision Letter 1]

9 Nov 2023

Social influences on delayed gratification in New Caledonian crows and Eurasian jays

PONE-D-23-21809R1

Dear Dr. Miller (Harrison),

We’re pleased to inform you that your manuscript has been judged scientifically suitable for publication and will be formally accepted for publication once it meets all outstanding technical requirements.

Kind regards,

Brenton G. Cooper, Ph.D.

Academic Editor

PLOS ONE
---

## [Editor Report · Acceptance letter]

16 Nov 2023

PONE-D-23-21809R1 

Social influences on delayed gratification in New Caledonian crows and Eurasian jays 

Dear Dr. Miller:

I'm pleased to inform you that your manuscript has been deemed suitable for publication in PLOS ONE. Congratulations! Your manuscript is now with our production department. 

Kind regards, 

on behalf of

Dr. Brenton G. Cooper 

Academic Editor

PLOS ONE